# When the Blood Hits Your Brain: The Neurotoxicity of Extravasated Blood

**DOI:** 10.3390/ijms22105132

**Published:** 2021-05-12

**Authors:** Jesse A. Stokum, Gregory J. Cannarsa, Aaron P. Wessell, Phelan Shea, Nicole Wenger, J. Marc Simard

**Affiliations:** 1Department of Neurosurgery, University of Maryland School of Medicine, Baltimore, MD 21201, USA; gcannarsa@som.umaryland.edu (G.J.C.); awessell@som.umaryland.edu (A.P.W.); pshea@som.umaryland.edu (P.S.); NMWenger@som.umaryland.edu (N.W.); MSimard@som.umaryland.edu (J.M.S.); 2Departments of Pathology and Physiology, University of Maryland School of Medicine, Baltimore, MD 21201, USA

**Keywords:** intracerebral hemorrhage, aneurysmal subarachnoid hemorrhage, neurotoxicity of blood, evacuation of CNS hemorrhage

## Abstract

Hemorrhage in the central nervous system (CNS), including intracerebral hemorrhage (ICH), intraventricular hemorrhage (IVH), and aneurysmal subarachnoid hemorrhage (aSAH), remains highly morbid. Trials of medical management for these conditions over recent decades have been largely unsuccessful in improving outcome and reducing mortality. Beyond its role in creating mass effect, the presence of extravasated blood in patients with CNS hemorrhage is generally overlooked. Since trials of surgical intervention to remove CNS hemorrhage have been generally unsuccessful, the potent neurotoxicity of blood is generally viewed as a basic scientific curiosity rather than a clinically meaningful factor. In this review, we evaluate the direct role of blood as a neurotoxin and its subsequent clinical relevance. We first describe the molecular mechanisms of blood neurotoxicity. We then evaluate the clinical literature that directly relates to the evacuation of CNS hemorrhage. We posit that the efficacy of clot removal is a critical factor in outcome following surgical intervention. Future interventions for CNS hemorrhage should be guided by the principle that blood is exquisitely toxic to the brain.

## 1. Introduction

Acute CNS hemorrhage, including intracerebral hemorrhage (ICH), intraventricular hemorrhage (IVH), subarachnoid hemorrhage (SAH), subdural hematoma (SDH), and traumatic contusion, comprises a large portion of the modern neurosurgical case load. While neurosurgical techniques have greatly expanded in recent decades, these diseases still leave most patients dead or highly dependent.

### 1.1. Intracerebral and Intraventricular Hemorrhage

ICH and IVH, which occur in ~45% cases of ICH [1], are major causes of morbidity and mortality worldwide. ICH comprises 10–20% of all strokes, with an incidence that ranges up to 200 people per 100,000 person-years [2,3]. New ICH cases have risen by nearly 50% between 1990 and 2010 [4]. Mortality after ICH is ~40% at 1 month and ~50% at 1 year [3]. ICH is highly morbid, with only 12–39% of patients achieving functional independence after 1 year [3]. Due to this morbidity and mortality, ICH is a very costly disease, with one Canadian study of 987 patients showing an average discharge cost of USD 10,544, and a maximum cost of USD 265,470 [5].

Efforts to improve ICH outcomes through changes in medical management have been unsuccessful. Intensive BP control (INTERACT-2, ATACH-2), administration of platelets (PATCH), recombinant factor VIIa (FAST), and tranexamic acid (TICH-2) have all failed to show substantial improvements [6,7,8,9,10].

In ICH, hematoma volume is the strongest predictor of death and function outcome at 1 month [11,12,13,14]. Thus, the major causative factor in the failure of the aforementioned trials may be the toxicity of the intracerebral blood.

### 1.2. Aneurysmal Subarachnoid Hemorrhage

Aneurysmal subarachnoid hemorrhage (aSAH) accounts for ~5–10% of all strokes in the US and has an admission rate of approximately 10 people per 100,000 person-years [15,16]. Aneurysm rupture accounts for approximately ~80% of spontaneous subarachnoid hemorrhage cases [17]. Like ICH, aSAH is highly morbid and mortal. Patients with aSAH suffer from a ~8–66% case mortality [18,19]. Approximately 25% of survivors are unable to return to functional independence. Furthermore, only 62% of survivors can return to any form of work, and only 33% are able to return to their previous job [20]. Hospitalization after aSAH is often long, with an average length of stay of ~20 days, and often expensive, with mean hospital charges of ~USD 170,000 [21]. In aSAH, both the initial clot volume and rate of clot clearance predict delayed cerebral ischemia, mortality, and severe disability at 3 months [22,23,24].

### 1.3. Extravasated Blood: More Than Just a Mass Lesion

Given the impact of extravasated blood, it is surprising that in many cases of intracranial hemorrhage, the extravasated blood is left in place for the neuroparenchyma to degrade and reabsorb. One US study reported that only 7% of all ICH patients were treated with surgical evacuation [25]. The bias against evacuation can be traced to several clinical trials that have yielded results that argue against routine evacuation of ICH. As a result, blood is often viewed simply as a cause of mass effect, and the decision to evacuate is made accordingly. In the present article, we posit that blood, rather than being merely a mass lesion, is exquisitely neurotoxic to the CNS. We first show that blood is composed of multiple, deleterious neurotoxins. Next, we explore the clinical evidence that supports the evacuation and clearance of CNS hemorrhage, and hypothesize why prior clinical trials have failed to show efficacy for clot evacuation.

## 2. Blood Harbors Numerous Neurotoxins

Blood harbors several direct neurotoxins, all of which are present at levels above that which can kill neurons (Table 1).

Blood also contains numerous substances that exacerbate morbidity after CNS hemorrhage. In the following section, we discuss the various blood neurotoxins, including their mechanism of action and their maladaptive effects (Table 2).

### 2.1. Thrombin

Thrombin is a serine protease that is primarily generated by the liver as the prothrombin proenzyme and is best known as a key factor in the coagulation cascade via generation of fibrin [96,97]. Circulating prothrombin is matured to thrombin at the site of activation of the coagulation cascade [97]. While circulating thrombin is normally excluded by the blood–brain barrier (BBB), disruption of the BBB after acute CNS injury results in extravasation. In addition, thrombin is generated at low levels by most CNS cells in the normal healthy brain [98].

Independent from its role in coagulation, thrombin also serves as an important signaling molecule. Thrombin triggers cellular signal transduction through activation of a unique family of transmembrane receptors—the protease associated receptors (PARs)—which are activated not by ligand binding, but by proteolytic cleavage of an exposed domain [99]. In the human CNS, PAR receptors are expressed by endothelium [100], neurons, microglia, and astrocytes [101]. PAR receptors are exquisitely sensitive to thrombin, with a half-maximal response achieved by only ~50 pM of thrombin [99].

After it crosses the BBB, thrombin triggers PAR-dependent neuroinflammation [35]. Thrombin mediates microglial activation and cytokine production [36,37], and astrocytic gliosis [38], cellular proliferation [102], and matrix metalloproteinase (MMP) production, which can mediate BBB breakdown [103]. In phagocytes, thrombin induces chemotaxis and secretion of proinflammatory cytokines [39].

Endothelial thrombin signaling is particularly pertinent to CNS hemorrhage. In brain endothelium, thrombin activates a transcriptional network that produces a pro-inflammatory and pro-angiogenic phenotype [40,41,44,45,46]. Endothelial cells that are exposed to thrombin respond with contraction and rounding [42] with downregulation and dysfunction of junctional proteins [43], and with increased angiogenesis and proliferation [104,105]. Together, these changes manifest as increased permeability to circulating plasma [41,42,43]

Thrombin has been associated with a variety of maladaptive secondary injury events after CNS hemorrhage. First, thrombin can directly mediate the death of CNS cells. After hemorrhage, thrombin triggers PAR-mediated apoptotic cell death of neurons [27,32,33,34], and contributes to perihematomal cell death and neuroparenchymal degeneration [106]. Second, thrombin induces contraction of vascular smooth muscle [107] and vasoconstriction of the cerebral vasculature [108], which may contribute to vasospasm following subarachnoid hemorrhage. In experimental subarachnoid hemorrhage, antagonism of thrombin reduces vasospasm and improves neurological outcome [49,109]. Third, thrombin is a major mediator of perihematomal vasogenic edema following ICH. Infusion of thrombin into brain tissue directly causes edema formation [110], whereas inhibition of thrombin with hirudin after experimental ICH decreases edema [111]. Fourth, thrombin has also been associated with hydrocephalus formation after IVH. Intraventricular thrombin injection results in PAR-1 dependent ventriculomegaly [50,51] and extravasation of plasma proteins, secondary to dysregulated cadherin expression [112]. Lastly, thrombin has pro-epileptogenic effects [52,53].

### 2.2. Fibrinogen

Fibrinogen is primarily produced by hepatocytes and is matured to fibrin at the site of coagulation by thrombin. Normally, fibrinogen is excluded by the BBB. Like thrombin, fibrinogen has a primary role as a pro-coagulant via its ability to cross-link platelets [58], and a secondary role as a modulator of neuroinflammation. Fibrinogen mediates microglial activation, ROS generation, and disease progression in the EAE model of multiple sclerosis [55]. Fibrinogen also triggers astrogliosis, astrocyte scar formation [56], and macrophage activation [57].

Fibrin contributes to neuroinflammation through two major mechanisms. Firstly, fibrin can directly interact with cell surface receptors. Upon conversion of fibrinogen to fibrin, an epitope is unmasked, which permits fibrin-mediated binding and activation of the microglial CD11b-CD18 integrin receptor, resulting in microglial activation [58,59]. Secondly, fibrin can serve as a carrier for cytokines, such as TGF-β [56].

### 2.3. Complement

The complement cascade is a key mediator of innate cellular immunity and has important adaptive and maladaptive roles after CNS hemorrhage. Complement proteins are primarily synthesized by hepatocytes [113], although most components of the complement cascade can also be synthesized by CNS cells [114]. After CNS hemorrhage, complement activity is increased in the neuroparenchyma [115,116]. Evolutionarily, complement may serve to mediate clearance of apoptotic cellular debris, via opsonization of damaged and altered self-ligands [60,117].

Complement is a potent mediator of neuroinflammation. Complement activity generates two pro-inflammatory anaphylatoxins: C3a and C5a. The anaphylatoxins, through their corresponding receptors (C3aR and C5aR) [118,119], stimulate phagocyte chemotaxis and transmigration, release of proinflammatory cytokines, and induction of oxidative burst [60]. C3 and C5 inhibition results in less microglial activation, neutrophil infiltration, and proinflammatory cytokine production after ICH [61,62].

Most importantly, complement is the primary mediator of a delayed form of vasogenic edema which occurs ~3 days after ICH. In delayed vasogenic edema, extravasated erythrocytes are lysed, spilling their edema-genic contents [63], as discussed in detail below. Complement mediates erythrocyte lysis at this time point [64] and is believed to initiate the formation of delayed vasogenic edema [61,65,66,67].

### 2.4. Clot-Associated Cell Debris, Leukocytes, and Platelets

The hematoma itself is comprised of millions of cells, including erythrocytes, white blood cells, and platelets, that are ultimately degraded and lysed, thereby spilling their contents and generating cellular debris. The immune system contains mechanisms to recognize damaged self-signals, so-called damage-associated molecular patterns (DAMPs), which are expressed by cellular debris. Notable DAMPs include the high mobility group box 1 (HMGB1) protein, S100 proteins, heat shock protein (HSP), and fibrinogen [120]. DAMPs are recognized and cleared by phagocytes via a family of Toll-like receptors (TLRs) [121]. After hemorrhage, TLR4 activity is upregulated, resulting in activation of phagocytes [122]. While controlled TLR activity is necessary for debris clearance, excessive activation has been associated with worsened neurological deficit and cerebral edema after brain hemorrhage [90].

Extravasated leukocytes worsen secondary injury following brain hemorrhage. In one interesting study, wild-type versus TLR4 knockout mice were submitted to experimental ICH with blood from either wild-type versus TLR4 knockout mice. Surprisingly, the wild-type mice injected with TLR4 knockout blood exhibited minimal deficits and fared better than the other groups [68], suggesting that immune signaling within the clot was a prime mediator of post-hemorrhage secondary injury. Indeed, circulating lymphocytes are critical mediators of neuroinflammation after ICH, with 30-day outcomes being linked to circulating lymphocyte counts [69]. Fingolimod, a sphingosine-1-phosphate modulator that depletes circulating lymphocytes, has been shown in animal and clinical studies to reduce circulating lymphocytes, reduce neuroinflammation and perihematomal edema, and improve neurological outcomes after ICH [70,71].

Extravasated platelets have also been shown to mediate secondary injury after ICH. First, during coagulation and clotting, platelets release TGF-β, a 25-kD cytokine [75]. In animal experiments, ventricular TGF-β can directly cause ventriculomegaly and communicating hydrocephalus [73,74]. After SAH in humans, TGF-β is more abundant in the CSF of patients with hydrocephalus, and in patients that require CSF diversion [123]. Second, during coagulation, platelets generate and secrete HMGB1 [72]. HMGB1, which normally binds to nuclear DNA [124], also serves as a DAMP, thereby activating phagocytic cells through TLR2 and TLR4 ligation and NF-κB activation [125,126]. Activated immune cells subsequently can directly kill neurons through MMP and ROS production [127].

### 2.5. Hemolysate

CNS hemorrhage results in erythrocyte extravasation. During clearance of erythrocytes and hemoglobin, numerous toxic products are generated, which greatly contribute to morbidity after hemorrhage. Importantly, hemoglobin degradation is a normally beneficial and evolutionarily adaptive process and should not be viewed as a purely maladaptive phenomenon. Rather, it is only when the system becomes overwhelmed do toxic side-effects begin to accrue.

Intact erythrocytes are not inherently toxic. However, upon lysis, toxic intracellular components, collectively referred to as hemolysate, are released. Hemolysis is temporally delayed since MAC-mediated hemolysis occurs mostly at ~3 days after hemorrhage [63,64]. Hemolysate causes oxidative stress, neuroinflammation, and a variety of subsequent secondary injuries [63,78]. Hemolysate can also directly induce vasospasm of cerebral arteries [128,129]. Extravasated erythrocytes can be directly phagocytized. However, erythrocyte-laden macrophages often release their internalized heme or iron [130]. In addition, they may be killed by the ingested hemoglobin, resulting in spillage of their toxic contents [130].

After hemolysis, tetrameric hemoglobin is released from erythrocytes. Whole blood contains approximately 2.5 mM of hemoglobin [28]. Released extracellular hemoglobin, which underlies the majority of hemolysate toxicity [131], then undergoes step-wise degradation, yielding toxic intermediates. Many of the hemoglobin breakdown products induce toxicity through generation of reactive oxygen species (ROS). ROS cause oxidation of cellular lipids, proteins, and DNA, resulting in programmed cell death [132].

Tetrameric hemoglobin either dissociates into dimers, which further decompose in the extracellular space, or is cleared by binding to haptoglobin, a circulating glycoprotein that is normally synthesized by the liver but can also be upregulated by glial cells after injury [133]. Hemoglobin binding results in a conformational change in haptoglobin, enabling its recognition and internalization via the CD163 receptor [82,134]. Interestingly, haptoglobin also slows auto-oxidation of hemoglobin and retards formation of toxic degradation products [135]. After hemorrhage, the binding capacity of haptoglobin is quickly overwhelmed [136].

Dimeric hemoglobin becomes oxidized to methemoglobin (Fe^3+^), which upon exposure to reactive oxygen species can become further oxidized to ferryl hemoglobin (Fe^4+^) [137]. Various modifications of the globin molecule can occur, including free radical formation and globin–globin crosslinks [137]. Hemoglobin derivatives spontaneously release their heme moiety, which becomes oxidized to hemin [28]. Extracellular hemin can be internalized by CNS cells either as a hemin–hemopexin complex via the low density lipoprotein receptor 1 (LRP1) [138,139], or as hemin alone through the heme carrier protein 1 (HCP1) [140]. Intracellular hemin is degraded by neuronal and glial heme oxygenase (HO)-1 or -2 in a NADPH-utilizing reaction to biliverdin, carbon monoxide, and free iron [28,141]. Biliverdin is processed to bilirubin, whereas the free iron is sequestered by ferritin [28,142], which is finally degraded to inert hemosiderin by lysosomes [28].

Nearly every stage of hemoglobin degradation is toxic. Hemoglobin causes apoptotic cell death of cultured CNS cells with an LD50 of 8 µM, which is substantially lower than the 2.5 mM of hemoglobin found in blood [29,30,76,77,78]. When injected in vivo, hemoglobin causes neuronal death [143], edema formation [63,79], focal epileptiform activity [80], and ventriculomegaly [81,82]. Oxyhemoglobin, the auto-oxidized form of hemoglobin, is also toxic to cultured CNS cells [83,84,85,86] via generation of hydroxy radicals produced during its degradation to methemoglobin [86]. Importantly, through a combination of ROS generation and NO scavenging [144], oxyhemoglobin is a potent arterial spasmogen [87], and in primate models induces vasospasm with potency equal to whole blood [88]. Methemoglobin is a TLR4 agonist and can induce microglial activation and generation of pro-inflammatory cytokines [89]. Modified fragments of globin are similarly toxic, and can cause inflammation, inter-endothelial gap formation, and reduced junctional resistance [137].

Hemin, the oxidized form of heme, is strongly associated with neuronal death. Once completely released from hemoglobin, degraded blood contains approximately 10 mM of hemin [28], which is approximately the LD50 for neurons [29,30]. Hemin is also associated with edema formation [79]. Hemin mediates toxicity through several mechanisms. Firstly, hemin can directly degrade hydrogen peroxide, thereby generating ROS and free radicals [28]. Secondly, hemin can intercalate into the plasmalemma and thereby sensitizes endothelial cells to ROS-mediated damage [145,146], resulting in lipid breakdown and impaired plasmalemmal function [147]. Thirdly, hemin can directly trigger TLR4 signaling and neuroinflammation [90]. Fourthly, heme induces endoplasmic reticulum and mitochondrial stress responses [148,149].

Hemin is degraded by HO into biliverdin, carbon monoxide (CO), and free iron, a reaction that constitutes the rate-limiting step of heme degradation. There exists conflicting data regarding the role of HO after CNS hemorrhage. In studies where HO is overexpressed after hemorrhage, investigators reported a protective role. This effect may be due to clearance of heme and generation of the antioxidants biliverdin and CO [150,151]. In other studies, HO activity was found to be maladaptive, with HO knockout resulting in reduced ROS generation and cell death [152,153]. These results may be due to an overload of ferritin binding capacity, and a depletion of intracellular NADPH, with subsequent depletion of glutathione [28]. The pleotropic roles of bilirubin and CO may also underlie the seemingly conflicting results. CO acts as a regulator of vascular tone, apoptosis and oxidative stress [154]. Bilirubin is a potent free radical scavenger that is also linked to encephalopathy.

Free iron itself is toxic. After ICH, free iron increases to a plateau at ~2 weeks and remains at elevated and toxic levels beyond 3 months after ictus [155,156]. Iron has been linked to dose-dependent CNS cell death [31,91,92,93,94], neuroinflammation [93], hydrocephalus [95], and edema formation [79,93]. Iron chelation is protective in many in vitro and in vivo models of CNS hemorrhage [30,78,79,157]. Iron mediates its toxic effects mostly through its ability to induce Fenton redox reactions with peroxides, resulting in production of ROS and free radicals [28]. In astrocytes, this process has been associated with loss of mitochondrial potential and ATP depletion [91].

## 3. Clinical Evidence Supporting Early and Total Evacuation/Treatment of Intracranial Hemorrhage

Having demonstrated the toxicity of blood, we now review the existing clinical literature on CNS hemorrhage with the hypothesis that interventions resulting in complete or near-complete evacuation and clearance of extravasated blood would be associated with improved outcomes following CNS hemorrhage. First, before examining the surgical literature, the medical management of ICH is reviewed and summarized in Table 3. Multiple RCTs for medical management, including varying levels of blood pressure control and multiple trials of pro-coagulant/pro-thrombotic agents, found no improvement in functional outcomes. The findings and analysis of medical managements of ICH trials have been summarized in a Cochrane Systematic Review last updated in 2018 [158]. Since no form of medical management addresses the complex neurotoxicity of CNS hemorrhage, the failure of these medical trials and future medical trials is not unexpected.

Next, a review of the surgical literature found many past trials of surgical evacuation of ICH as well as multiple systematic reviews and meta-analyses of those trials.

Critical to our analysis of the existing literature was to determine how effective these groups were in removing the blood clot, as our hypothesis is that those groups that removed the most blood would have better outcomes. However, a large majority of previous trials and retrospective studies did not include any determination or measurement of hematoma evacuation/clearance efficacy (Table 4). Likewise, the systematic reviews and meta-analyses that have been conducted on these trials suffer from the same issue of not having any measure or determination of hematoma evacuation efficacy within their analysis. Instead, they grossly compare the outcomes of surgical intervention versus non-intervention without first examining the efficacy of the intervention [162,163,164]. Not surprisingly, these systematic reviews have ambivalent conclusions about the efficacy of surgical intervention.

### 3.1. Intracerebral Hemorrhage: Evacuation Efficacy

The existing literature on the surgical evacuation of ICH possesses several weaknesses in study design, including a lack of reporting of the extent of clot evacuation, inconsistency in the timing of the intervention, and the heterogeneity of surgical approaches of evacuation. Notably, few studies have reported on the extent of hematoma evacuation, which we posit to be a critical determinant of surgical intervention outcomes. Hematoma evacuation efficacy (HEE) is defined in Equation (1):*HEE = ((Post-operative ICH volume/Pre-operative ICH volume)* × 100%)(1)

Akin to the literature for thrombectomy for ischemic stroke, which found no benefit for thrombectomy with low recanalization rates, low HEE may suggest the need for better techniques rather than abandonment of surgical intervention altogether. Large randomized trials of ICH evacuation including STICH and STICH II found no improvement in outcome between surgical intervention and medical management [173,181]. However, neither study measured HEE. MISTIE III, a randomized trial of ICH evacuation via catheter-based thrombolysis, did measure hematoma evacuation rate and extent. While MISTIE III was not significant for its primary outcome, subsequent post hoc analyses found a significant increase in good functional outcome (mRS 0–3) in patients with final hematoma volume <15 cc compared to those with ≥15 cc hemorrhage (53.1% vs. 32.7%). Furthermore, patients with volume <15 cc also had outcomes significantly improved compared to the standard medical therapy group [188,189,190]. Another post hoc analysis of MISTIE III reported significant improvement in functional outcome in patients with end-of-treatment blood volume <5 cc compared to 5–10, 10–20, and >20 cc [191]. In the last study, the 30 patients in the <5 cc group had a 73.3% rate of good functional outcome (mRS 0–3), while the >20 cc group had a 28.1% rate of good functional outcome. Evacuating another 15 cc of blood (the difference between these two groups) resulted in 45.2% absolute change in rate of good functional outcome. Thus, while MISTIE III failed to show a benefit for surgical intervention, we posit that the subsequent post hoc analyses offer the strongest evidence to date that high HEE leads to dramatic improvement in functional outcomes.

In the smaller ICH evacuation studies that report HEE, there is a trend towards improved outcomes in the studies with the highest evacuation rates. Gross total hematoma removal has rarely been attempted in ICH, with a single series of 176 patients showing faster improvement in functional status and decreased level of CSF neuroinflammatory markers compared to subtotal hematoma removal [192]. In contrast, a report on stereotactic thrombolysis of ICH, which recorded low (10–20%) evacuation rates, failed to show any functional or mortality benefit with surgery [171]. Multiple reports of endoscopic ICH evacuation with recorded evacuation rates of 70–80% have shown improved functional outcomes and a mortality benefit in surgically treated patients [166,176,183].

### 3.2. Intracerebral Hemorrhage: Surgical Timing

Many of the previously completed trials on surgical evacuation of ICH have primarily focused on non-emergent evacuation of the hemorrhage, with several patients being randomized to treatment or, less frequently undergoing treatment, within a range of 24–72 h following the ictus (Table 4). Based on current understanding of the pathophysiology of ICH and the ensuing secondary brain injury, these studies may have been set up for failure by nature of their design and the timing of surgical intervention.

Secondary injury refers to injury resulting from peri-hemorrhagic inflammation, toxic blood breakdown products, and perihematomal edema (PHE) that occurs following hemorrhage, which can contribute to neurologic deterioration several hours to days following the ictus [193]. PHE specifically increases most rapidly in the first two days following hemorrhage onset and can develop for up to two weeks [194]. PHE volume is directly related to the initial ICH volume at all times after the ictus and the rate of PHE expansion in the 72 h after the ictus is associated with poor functional outcomes in patients with deep ICH [193,195].

Conceptually, it would seem most beneficial to evacuate ICH as early as possible to minimize the cascade of secondary injury and evolution of PHE. This concept is supported by animal models of ICH evacuation that have found that evacuation was most beneficial in terms of minimizing permanent neurologic deficit if performed within 6 h of onset. Evacuation was still beneficial, although to a lesser extent, if performed within 12 h of onset. Animals evacuated earlier had reduced brain edema formation, reduction in blood–brain barrier disruption, and reduced perihematomal glutamate content, suggesting less secondary injury [196,197]. Analysis of MISTIE II evacuation showed surgical reduction in hematoma volume was associated with a parallel reduction in PHE volume [198].

The failure of large ICH evacuation trials including STICH I and STICH II may relate to the delay in surgical intervention. In these studies, the average times to surgery were 30 and 26.7 h for STICH I and STICH II, respectively [173,181]. Another comparison to the stroke thrombectomy literature may be appropriate, since in that literature, very few patients ultimately benefit from intervention after 6 h. While the pathophysiology of ICH and ischemic stroke is quite different, the notion that “time is brain” may apply in both cases.

The study of timing of ICH evacuation in humans is extremely limited. Additionally, in studies that did report timing, very few also reported HEE. A randomized clinical trial by Wang et al. assessed the impact of surgical timing on 500 study subjects randomized to either surgical evacuation or medical therapy alone [178]. Perioperative and long-term functional outcomes, in addition to mortality, were improved in surgically treated patients relative to medically treated controls if surgery was performed in an ultra-early (≤7 h from onset) or early (7–24 h) timeframe. Patients treated surgically beyond 24 h from onset did not experience any benefit in terms of functional outcome or mortality. Of note, the authors encourage exercising caution when operating in the ultra-early period due to an increased risk of re-hemorrhage [178].

In 1999, Zuccarello and colleagues called for the initiation of a randomized clinical trial of early or ultra-early ICH evacuation (within 3–6 h of onset) [170]. To date, no such study has been completed. However, the establishment of systems of care dedicated to rapid triage and management for acute thrombectomy for ischemic stroke over the past decade now make such a study feasible. Early surgical evacuation of ICH, if performed in a minimally invasive fashion, may have the potential to minimize secondary injury, reduce edema and mass effect, and improve clinical outcomes. The future completion of such a study will be important for the advancement of ICH care and may offer the most promise in terms of improving patient outcomes.

### 3.3. Intracerebral Hemorrhage: Method of Intervention

Heterogeneity of surgical approaches to hematoma evacuation has plagued the ICH literature. Methods of surgical intervention for ICH previously reported include: decompressive craniectomy (DC) alone, DC + ICH evacuation, craniotomy (size not consistently reported) for evacuation, burr holes with stereotactic placement of drainage catheters, endoscopic-guided drainage of ICH, drainage using the Apollo suction system, and, most recently, image-guided minimally invasive hematoma evacuation using the BrainPath (NICO Corporation) system.

Consistent with the concept that blood is toxic to the brain and that it should be the primary target of any ICH intervention, DC alone has not been shown to improve functional outcomes. A systematic review and meta-analysis examining the efficacy of DC in ICH found no association with improved functional outcomes but did show a reduction in mortality rate, likely due to relief of mass effect [199]. The only RCT within the meta-analysis was a study of 40 patients comparing DC + ICH evacuation to ICH evacuation alone in patients with large (>60 cc) ICH, which showed a significant increase in favorable outcome in the DC + ICH group compared to the ICH evacuation group [200]. This study reported many baseline and outcome data points but did not report HEE. In contrast, a small retrospective series comparing ICH evacuation with quantified HEE (92% average evacuation) vs. ICH evacuation + DC found no benefit for the addition of DC in terms of functional outcome, suggesting near-complete hematoma evacuation alone is sufficient for treatment of ICH. Further, a multi-variate analysis showed post-operative hematoma volume ≤2 mL to be a significant predictor of good functional outcome, supporting the idea that gross total evacuation of ICH, rather than relief of mass effect, is the critical factor to improving outcomes [201].

Recent reports using the BrainPath system (NICO Corporation) have found encouraging results with high reported hematoma evacuation rates of 92% and 96% in two small series of 16 and 18 patients, respectively [202,203]. In a larger series of 39 patients treated with the BrainPath technique of hematoma evacuation, >90% evacuation was achieved in 72% of patients [204]. The ongoing ENRICH (Early miNimally invasive Removal of IntraCerebral Hemorrhage) trial will offer further insight into the efficacy of this technique. An advantage of the BrainPath technique is not only its minimally invasive nature, but also its attention to avoidance of critical white matter tracts that may be overlying or adjacent to the ICH by use of reliable operative corridors to avoid these important regions.

### 3.4. Intraventricular Hemorrhage

ICH is often accompanied by bleeding into the cerebral ventricles, or intraventricular hemorrhage (IVH). IVH is associated with up to 50% mortality after ICH, and can also occur with SAH, traumatic hemorrhage, or as an isolated hemorrhage [205]. In addition to predicting mortality, IVH is correlated with aseptic CSF inflammation [206].

Trials of IVH clearance support the notion that effective blood clearance improves outcomes, although they suffer from similar issues as the ICH trials including long periods from ictus to intervention and low clearance rates. The CLEAR trial examined the effect of IVH removal with intraventricular TPA (IVTPA) upon functional outcomes and mortality [207]. In this study, randomization occurred at 52 h with first IVTPA treatment given at 55 h on average. Thirty-three percent of patients in the IVTPA group achieved 80% clearance vs. 10% in the saline group. At 180 days, the IVTPA group had significantly lower mortality compared to control (18% vs. 29%), although this group did not demonstrate improved functional outcomes when compared to control. Higher absolute clearance rates are likely to be necessary to have a greater impact on outcomes. This is also suggested by a post hoc analysis of MISTIE III, which, while primarily focused on clearance of ICH, also showed that lower end-of-treatment IVH volumes are associated with significant improvements in good outcome [191]. Those patients with EOT IVH volumes of >3 mL had 15% likelihood of achieving mRS 0–3 compared to 70% likelihood of achieving mRS 0–3 for those with EOT IVH volumes of 0 mL.

The most direct, immediate method of IVH clearance that has been studied to date is endoscopically guided evacuation. Multiple small trials have shown favorable outcomes for endoscopic clearance of IVH when compared to EVD alone or EVD + tPA. In a 48-patient randomized trial comparing endoscopic IVH evacuation compared to EVD alone, the endoscopic evacuation group was found to have significantly lower shunt-dependent hydrocephalus with shorter ICU stays compared with EVD alone [208]. Another randomized trial of endoscopic IVH evacuation vs. EVD placement in the setting of thalamic ICH found endoscopic surgery to be associated with a lower GOS score and lower onset rates for shunt-dependent hydrocephalus and aspiration-related pneumonia in comparison to EVD [209]. High evacuation rate was associated with lower shunt-dependent rate and shorter hospitalization. A meta-analysis comparing endoscopic evacuation of IVH in the setting of ICH as compared to EVD + intraventricular fibrinolytics found an association of lower mortality (OR 0.31), higher effective hematoma evacuation rate (OR 25.50), higher likelihood of good functional outcome (OR 4.51), and lower rate of shunt dependence (OR 0.16) [210].

Lumbar drainage has also been examined as a mode of IVH clearance, either alone or as an adjunct to IVTPA. A prospective pilot trial in patients with IVH found that IVTPA + lumbar drain was associated with dramatically decreased shunt dependence [211]. A follow-up randomized trial by the same group examined use of IVTPA + lumbar drain compared to IVTPA alone for reducing the likelihood of shunt dependence [212]. The trial was stopped early due to the efficacy of the intervention. There were no patients (0/14) in the IVTPA + lumbar drain group who ultimately required shunt placement compared to 43% of patients (7/16) in the control group of IVTPA alone. Further, bleeding complications related to IVTPA administration were significantly less likely in the IVTPA + lumbar drain group (OR 0.401).

### 3.5. Aneurysmal Subarachnoid Hemorrhage

If the presumption that the toxicity of blood is the primary mediator of the deleterious effects of aSAH, then it should hold that the amount and quantity of aSAH blood should be predictive of DCI, vasospasm, and poor outcomes. Indeed, radiological scales of the quantity of subarachnoid blood, such as the Hijdra scale, are strongly predictive of DCI/vasospasm and functional outcome [213,214]. Volumetric quantification of subarachnoid blood also reliably predicts DCI (adjusted OR of 1.02 for every 1 mL of subarachnoid blood) [215]. A prospective, comparative study of volumetric quantification against traditional qualitative scales of SAH blood volumes showed higher AUC and discriminative ability to predict DCI and outcome [216]. The radiological density of subarachnoid blood, another proxy for blood burden, is also predictive of vasospasm [217]. In addition to being predictors of DCI, scales of blood burden, including mFS, independently predict outcome in poor-grade subarachnoid hemorrhage [218,219]. Highlighting a potential role for thrombin in vasospasm, aSAH patients with greater blood burden had higher CSF thrombin activity, slower clearance of blood, and increased vasospasm [220].

The above clinical data, in combination with the known toxicity of blood previously demonstrated, suggest that the rapid and complete clearance of toxic subarachnoid blood, through either direct clearance or indirect means, may reduce risk of DCI, vasospasm, and DIND and consequently improve outcome.

In aSAH patients with anterior communicating aneurysms requiring open aneurysmal clipping, cisternal clot evacuation along with fenestration of lamina terminalis was associated with decreased mortality from vasospasm and DIND [221]. Small trials of aSAH clearance via other direct clearance methods have shown decreased rates of DCI and improved outcomes. A small, matched cohort trial of direct cisternal lavage with urokinase with the objective of SAH clearance showed decreased rates of DCI and mortality with a trend towards decreased shunt dependence and improvement in functional outcomes at time of rehabilitation discharge [222]. The trial did not assess for efficacy of SAH clearance but mentioned that 9 of 20 patients with early CTs exhibited near complete resolution of SAH. Increasing dosages of urokinase leading to increased clearance of drained blood were associated with less vasospasm in one study comparing differing dosages of urokinase [223]. Consecutive series of aSAH patients undergoing a similar cisternal irrigation method showed dramatically lower rate of vasospasm and DIND [224]. Intraoperative and post-operative cisternal irrigation with tPA has also been shown to be safe and effective for prevention of vasospasm [225]. One interesting study indicated that head shaking in conjunction to cisternal irrigation improved rates of vasospasm and DIND, presumably due to improved mobilization of clot [223]. Drainage of the cisterns does not seem to improve outcomes in patients with thin SAH, consistent with the theory that removing the toxic blood is the mechanism by which this intervention works [226].

Indirect clearance of aSAH via ventricular or lumbar drainage of CSF has also been shown to decrease rates of DCI/vasospasm and improve functional outcomes in multiple retrospective and prospective trials. A study that examined rates of symptomatic vasospasm in aSAH patients treated with external ventricular drain (EVD) compared to those who did not require EVD (NEVD) found significantly higher rates of vasospasm in the NEVD group despite the NEVD group having a much a higher population of low Hunt–Hess scale patients [227]. A small comparative study of ventricular drainage (EVD) vs. lumbar drainage patients showed that patients treated with lumbar drain had more rapid clearance of blood and significantly less new hypodense areas on CT compared to the EVD group, although both groups had similar rates of symptomatic vasospasm [228].

Multiple reports of lumbar drainage for aneurysmal subarachnoid hemorrhage have shown a decrease in DIND/vasospasm. Lumbar drainage has also been associated with shorter hospital stays and improved outcomes. The proposed mechanism involves increased clearance of cranial cisternal blood into the spinal subarachnoid space and lumbar cistern [229,230,231]. A retrospective study on 167 patients showed that lumbar drainage after SAH markedly reduces the risk of clinically evident vasospasm and its sequelae, shortens hospital stay, and improves outcome [229]. A prospective, randomized trial of 60 HH II-IV SAH patients showed that lumbar CSF drainage reduced clinical and radiographic vasospasm and also showed improved outcomes and a trend toward a shorter hospital stay [230]. A systematic review of eight studies with 841 patients showed that lumbar drainage was associated with reduced rates of symptomatic vasospasm and improved clinical outcomes in patients with good clinical grade with thick clot burden [231]. The LUMAS trial of lumbar drainage examined 210 people of varying grades of aSAH and showed reduced rates of delayed ischemic neurological deficit and improved early outcomes with lumbar drainage [232]. A dual-lumen lumbar drainage system (Neurapheresis^TM^) has been proposed as an alternative to simple lumbar drainage [233]. A recent clinical trial (PILLAR) has demonstrated safety and efficacy of this system, although further study is needed [234].

## 4. Conclusions

Intracranial blood, whether in the brain parenchyma, subarachnoid space, or ventricles, is toxic. The toxicity of blood in the CNS is related to its many components and breakdown products, which are harmful through a myriad of biological pathways. The past literature on medical management of intracranial hemorrhage is a history of failed trials, as no medical management conceived of thus far can adequately target all of the toxic pathways of blood and its breakdown products. The surgical literature of CNS hemorrhage evacuation and clearance, while also possessing a history of non-significant trials, has a uniting theme throughout: those trials that have tracked hematoma evacuation efficacy, whether in ICH, IVH, or aSAH, have found that high rates of HEE, or gross total evacuation, lead to improvements in functional outcomes. While surgical risk in clearing cerebral hemorrhage remains an important consideration, particularly in cases of deep hemorrhage, blood clearance via minimally invasive surgical evacuation, lumbar drainage, direct lavage and other modalities all hold promise for improving outcomes and decreasing the formidable morbidity of intracranial hemorrhages. The future of ICH, IVH, and aSAH research and management should be dedicated to the first principle that blood is toxic to the brain.

## Figures and Tables

**Table 1 ijms-22-05132-t001:** Blood components with direct neurotoxic effects, their concentration in whole blood, the concentration needed to achieve half-maximal neuronal death (LD50) in cultured neurons, and their blood concentration relative to their LD_50_.

Blood Component	Concentration in Whole Blood	LD_50_ in Neurons at 24 h	Blood Concentration Relative to LD_50_
Thrombin	30 U/mL [26]	~4 U/mL [27]	7.5x
Hemoglobin	2.5 mM [28]	1–8 µM [29,30]	~1000x
Free Iron	10–30 µM	~10 µM [31]	1–3x

**Table 2 ijms-22-05132-t002:** Blood neurotoxins and their individual modes of toxicity.

Component	Deleterious Effect	References
Thrombin	Neuron Death	[27,32,33,34]
	Neuroinflammation	[35,36,37,38,39,40,41]
	Cerebral Edema	[41,42,43,44,45,46,47,48]
	Vasospasm	[48,49]
	Hydrocephalus	[50,51]
	Seizure	[52,53,54]
Fibrinogen	Neuroinflammation	[55,56,57,58,59]
Complement	Neuroinflammation	[60,61,62]
	Cerebral Edema	[61,63,64,65,66,67]
Leukocytes	Neuroinflammation	[68,69,70,71]
Platelets	Neuroinflammation	[72]
	Hydrocephalus	[73,74,75]
Hemoglobin	Neuron Death	[29,30,76,77,78]
	Cerebral Edema	[63,79]
	Seizure	[80]
	Hydrocephalus	[81,82]
Oxyhemoglobin	Neuron Death	[83,84,85,86]
	Vasospasm	[87,88]
Methemoglobin	Neuroinflammation	[89]
Hemin	Neuron Death	[29,30]
	Cerebral Edema	[79]
	Neuroinflammation	[90]
Iron	Neuron Death	[31,91,92,93,94]
	Cerebral Edema	[79,93]
	Neuroinflammation	[93]
	Hydrocephalus	[95]

**Table 3 ijms-22-05132-t003:** Summary table of the major RCTs for medical management of intracerebral hemorrhage.

Study	Year Published	Intervention	Outcome
FAST [159]	2008	Factor VII vs. placebo	No difference in outcome
INTERACT-2 [7]	2013	BP < 140 vs. BP < 180 mm Hg	No difference in outcome
ATACH-2 [6]	2016	BP 110–139 vs. 140–179 mm Hg	No difference in outcome
PATCH [8]	2016	Platelet transfusion vs. standard care	Worse outcome in transfusion group
TICH-2 [10]	2018	Tranexamic Acid vs. placebo	No difference in outcome
STOP-AUST [160]	2020	Tranexamic Acid vs. placebo for spot sign ICH	No difference in outcome
I-DEF [161]	2020	Deferoxamine Mesylate vs. placebo	No difference in outcome

**Table 4 ijms-22-05132-t004:** Summary of clinical literature evaluating the potential benefit of traditional open surgery and/or minimally invasive surgery for evacuation of intracerebral hemorrhage.

Study	Number of Study Subjects (Surgery; Control)	Time from Onset to Randomization * or Treatment	Hematoma Evacuation Efficacy Measured?	Average Rate of Hematoma Evacuated in Intervention Group	Primary Surgical Technique	Functional Benefit?	Mortality Benefit?
McKissock et al. (1961) [165]	*n* = 180 (89 surgery, 91 control)	72 h	No	NA	Open Surgery	No	No
Auer et al. (1989) [166]	*n* = 100 (50 surgery; 50 control)	48 h	Yes	Est. Avg.: 71% 50–70%: 56% 70–90%: 29% 90–100%: 15%	Endoscopic Surgery	Yes	Yes
Juvela et al. (1989) [167]	*n* = 52 (56 surgery; 56 control)	48 h	No	NA	Open Surgery	No	Yes
Batjer et al. (1990) [168]	*n* = 21 (8 surgery; 13 control)	24 h *	No	NA	Open Surgery	No	No
Morgenstern et al. (1998) [169]	*n* = 34 (17 surgery; 13 control)	Avg: 1.2 h * (surgery) Avg: 5.1 h * (control)	No	NA	Open Surgery	No	No
Zuccarello et al. (1999) [170]	*n* = 20 (9 surgery; 11 control)	Avg: 8 h and 35 min	No	NA	Open Surgery/Stereotactic aspiration (deep hemorrhages)	No (GOS or Rankin Scale); lower follow-up NIHSS in surgical group	No
Teernstra et al. (2003) [171]	*n* = 70 (36 surgery; 34 control)	72 h	Yes	10–20%	Stereotactic thrombolysis	No	No
Hattori et al. (2004) [172]	*n* = 242 (121 surgery; 121 control)	24 h	No	NA	Stereotactic	Yes	Yes
Mendelow et al. (2005) [173]	*n* = 964 (468 surgery; 496 control)	<72 h * (Avg. time from ictus to surgery: surgery 30 h; control 60 h ¥)	No	NA	Open Surgery	No	No
Pantazis et al. (2006) [174]	*n* = 108 (54 surgery; 54 control)	8 h	No	NA	Open Surgery (15–20 mm dural incision)	Yes	No
Wang et al. (2008) [175]	*n* = 500 (266 surgery; 234 control)	≤7 h 7–24 h Or >24 h	No	NA	MIS, endoscopic, stereotactic, open surgery	Yes (those treated <24 h)	Yes (those treated <24 h)
Miller et al. (2008) [176]	*n* = 10 (6 surgery; 4 control)	24 h *	Yes	Avg. 80%	Endoscopic	NA	NA
Kim et al. (2009) [177]	*n* = 387 (284 surgery; 183 control)	5 days	No	NA	Stereotactic	Yes	No
Wang et al. (2009) [178]	*n* = 377 (195 surgery; 182 control)	Avg. 21 h	No	NA	Stereotactic	Yes	No
Sun et al. (2010) [179]	*n* = 304 (159 MIS; 145 control ¥)	72 h *	No	NA	Stereotactic	Yes	No
Zhou et al. (2011) [180]	*n* = 168 (90 MIS; 78 control ¥)	24 h *	No	NA	Stereotactic	Yes	No
Mendelow et al. (2013) [181]	*n* = 601 (307 surgery; 294 control)	48 h *	No	NA	Open surgery	No	Yes (in superficial hematoma subgroup)
Zhang et al. (2014) [182]	*n* = 31 (21 MIS; 30 control ¥)	24 h.	No	NA	Endoscopic surgery	Yes	No
Feng et al. (2016) [183]	*n* = 184 (93 MIS; 91 control ¥)	24 h *	Yes	≥70%	Endoscopic surgery	Yes	NA
Hanley et al. (2016) [184]	*n* = 96 (54 surgery; 42 control)	48 h *	No	NA	Stereotactic	No	No
Vespa et al. (2016) [185]	*n* = 56 (14 surgery; 42 control ¥¥)	48 h *	Yes	Avg. 68%	Endoscopic	Yes	No
Yang and Shao (2016) [186]	*n* = 156 (78 surgery, 78 control)	Unknown	Yes	75%	Craniopuncture	Yes	NA
Hanley et al. (2019) [187]	*n* = 506 (255 surgical, 251 control)	36 h *	Yes	NA	Stereotactic	No	Yes

Abbreviations: NA, not applicable; MIS, minimally invasive surgery; GOS, Glasgow Outcomes Scale; NIHSS, National Institutes of Health Stroke Scale. ¥ represents studies of MIS compared to a control group of patients treated with open surgery. ¥¥ includes 36 patients comprising a contemporaneous medical control group from the MISTIE trial.

## Data Availability

Not applicable.

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
