# Peer review of "When the Blood Hits Your Brain: The Neurotoxicity of Extravasated Blood"

_ijms, 2021, doi:10.3390/ijms22105132_

Round 1

Reviewer 1 Report

This paper is a comprehensive review on the potent neurotoxicity of blood after different patterns of cerebral hemorrhage such as IVH and SAH.

After an excursus on the pathogenetic cascade that is triggered after the different forms of cerebral hemorrhage, the authors reviewed the previous RTCs concerning the medical and surgical therapy of the various forms of cerebral hemorrhage, underlining what that is largely known i.e. that many of them did not show significant benefit of clot removal on functional prognosis.
The authors therefore mainly based their criticism on the fact that these studies did not precisely quantize the amount of extravasated blood that was removed by surgery.
If on the one hand the presence of extravasated blood and its catabolic products can trigger numerous pathogenetic mechanisms, on the other hand the state of the art of the evidence dictated by these RTCs is that surgery does not change the prognosis in most cases. Furthermore, the authors completely neglect the consideration that the choice to evacuate a cerebral hemorrhage must always be balanced with the surgical risks, for example in the case of deep hemorrhages.

Therefore, I suggest to moderate the final message of this review, limiting these considerations to those cases in which the balance between the surgical evacuation and the best medical treatment is largely in favor of the former as for in case of a superficial hematoma.
Furthermore, the manuscript is extremely long and needs to be greatly shortened.

Reviewer 2 Report

The paper by Stokum et al is of potential interest, yet comes with severe issues. The amount of data presented is overwhelming and poorly structured. Moreover experimental, preclinical and clinical data, as well as causation and associations are mixed only to serve one purpose: to convince the readers that extravasated blood is toxic and contributes to the sequalae of intracerebral hemorrhage.

Upon reading and rereading the paper I was left with just one thought: what was the reason for not performing a proper meta-analysis or a Cochrane review of the evacuation trials? If this review provides a supportive signal, a review of the potential molecular pathways would be justified. The data presented in the paper could serve as an outline, yet this review would benefit from a more explanatory style of writing. (some examples: TLR4 KO blood, are we considering a soluble factor or is it a leucocyte-mediated effect, the complex regulation of thrombin activity that requires activation but is also mitigated by specific and generic inhibitors (alph2M), transport of haptoglobin and hemopexin over the blood brain barrier). Unfortunately, biology is not linear...... 

Author Response

The paper by Stokum et al is of potential interest, yet comes with severe issues. The amount of data presented is overwhelming and poorly structured. Moreover experimental, preclinical and clinical data, as well as causation and associations are mixed only to serve one purpose: to convince the readers that extravasated blood is toxic and contributes to the sequalae of intracerebral hemorrhage.

Upon reading and rereading the paper I was left with just one thought: what was the reason for not performing a proper meta-analysis or a Cochrane review of the evacuation trials? If this review provides a supportive signal, a review of the potential molecular pathways would be justified. The data presented in the paper could serve as an outline, yet this review would benefit from a more explanatory style of writing. (some examples: TLR4 KO blood, are we considering a soluble factor or is it a leucocyte-mediated effect, the complex regulation of thrombin activity that requires activation but is also mitigated by specific and generic inhibitors (alph2M), transport of haptoglobin and hemopexin over the blood brain barrier). Unfortunately, biology is not linear...... 

  • Thank you for your valuable feedback. It is understandable to question why we did not perform a systematic review/meta-analysis of the existing clot evacuation literature. We have included edits and new references to elaborate our reasoning, in an updated section at line 300. Specifically, systematic reviews and meta-analyses of clot evacuation literature do not account for hematoma evacuation efficacy (HEE). We have included references to multiple systematic reviews and meta-analyses of clot evacuation which have already been performed, but do not account for HEE. This is a principal argument of our manuscript - that the literature to date has failed to focus on examining the efficacy of surgical clot (blood) evacuation. The difficulty in performing a comprehensive analysis of prior ICH evacuation trials is that most well-known RCTs, such as STICH I and STICH II, would have to be excluded as they did not measure hematoma evacuation efficacy in their analysis.
  •     Ultimately, it is our strong belief that the review format is most suitable to illustrate our principal argument that blood is toxic to the brain, as a systematic review of clinical literature would have to exclude examining the basic science literature as we are able to include in the review format. Certainly, this type of review is atypical, but we feel part of the value of our contribution to the literature is in the structure of this review, which aims to comprehensively examine and posit the toxicity of blood to the brain from both a basic science and clinical perspective.

Reviewer 3 Report

The paper represents an important summary and review of clinical and pathophysiological characteristics of brain hemorrhage. The authors collect important knowledge for both clinicians and basic scientists. The language is correct, readable, yet concise. There is hardly any spelling in it. It describes almost all clinical studies in the field of stroke, including the results of recent publications. The content of the article meets the expectations of a systematic review, although it is not, however, the information provided is very important and accurate. The literature used is critically selected, praiseworthy for using data from original works rather than repeating statement from other review papers.

The next important conclusion is found in the abstract: “Trials of medical management for these conditions over the past decades have been largely unsuccessful in improving outcome and reducing mortality”. The aim of this paper is based on the abstract: “In this review, we evaluate the direct role of blood as a neurotoxin and its subsequent clinical relevance”.

The second important message of the paper is the beneficial effects of the effective blood clearance by surgical intervention on outcomes of different types of central nervous system hemorrhages. Two examples of the effectiveness of blood removal: “Multiple small trials have shown favorable outcomes for endoscopic clearance of IVH when compared to EVD alone or EVD+tPA”. “There were no patients (0/14) in the IVTPA+lumbar drain group who ultimately required shunt placement compared to 43% of patients (7/16) in the control group of IVTPA alone.” The postictal timing of surgical procedure is also important: “Early surgical evacuation of ICH, if performed in a minimally-invasive fashion, may have the potential to minimize secondary injury, reduce edema and mass effect, and improve clinical outcomes”.

These data underline the importance of the detrimental effects of extravascular blood.

The role of red blood cells and their constituents is extremely underrepresented in this review. Basic literature is missing.

  1. For example: the first papers about heme toxicity towards vascular endothelial cells sensitizing them against reactive oxygen species are very important publications opening the field of heme toxicity.
  2. In this review there is no mention about endogenous, intracellular protective systems against heme stress mediated free radical toxicity, for instance heme oxygenase-ferritin system, where ferritin is the ultimate protectant.
  3. New literatures discuss the toxic heme effect on endoplasmic reticulum and mitochondrial function, they should be included into the paper.
  4. The role of HO-1 should be discussed in a more detail form, as it is known as a controversial enzyme in the central nervous system. What is the role of bilirubin, the endproduct of the heme oxygenase-biliverdin reductase enzymes, after brain hemorrhage, is it a protective antioxidant or the agent of bilirubin encephalopathy.
  5. Carbon monoxide, another endproduct of HO-1 enzyme has multiple effects on cellular metabolism. It would be important to discuss in this article.
  6. In the field of hemoglobin, it needs to be clarified what forms of hemoglobin the authors mean by oxidized hemoglobin: ferro-hemoglobin with oxygen on it, methemoglobin, ferrylhemoglobin, hemichrome. They have different pathological effects.
  7. Table 1. Heme concentration derives from hemoglobin concentration in the whole blood. Only free heme has clinical relevance. Although the table originates from another paper, it is misleading. Since this not a clear table that is why it could lead to a not correct sentence: „Hemin, the oxidized form of heme, is strongly associated with neuronal death. Blood contains approximately 10 mM of hemin, which is approximately the LD50 for neurons.” This paragraph needs correction.
  8. It is not correct, that globin is an innocent player in hemorrhage. For example, peptides derived from oxidized hemoglobin represent proinflammatory effects.

Since the review states: “we evaluate the direct role of blood as a neurotoxin“, the above items should implement into this paper. Although the clinical study part of the article is based on uptudate literature, the situation not the same for the hemolysate paragraph.

Author Response

The paper represents an important summary and review of clinical and pathophysiological characteristics of brain hemorrhage. The authors collect important knowledge for both clinicians and basic scientists. The language is correct, readable, yet concise. There is hardly any spelling in it. It describes almost all clinical studies in the field of stroke, including the results of recent publications. The content of the article meets the expectations of a systematic review, although it is not, however, the information provided is very important and accurate. The literature used is critically selected, praiseworthy for using data from original works rather than repeating statement from other review papers.

The next important conclusion is found in the abstract: “Trials of medical management for these conditions over the past decades have been largely unsuccessful in improving outcome and reducing mortality”. The aim of this paper is based on the abstract: “In this review, we evaluate the direct role of blood as a neurotoxin and its subsequent clinical relevance”.

The second important message of the paper is the beneficial effects of the effective blood clearance by surgical intervention on outcomes of different types of central nervous system hemorrhages. Two examples of the effectiveness of blood removal: “Multiple small trials have shown favorable outcomes for endoscopic clearance of IVH when compared to EVD alone or EVD+tPA”. “There were no patients (0/14) in the IVTPA+lumbar drain group who ultimately required shunt placement compared to 43% of patients (7/16) in the control group of IVTPA alone.” The postictal timing of surgical procedure is also important: “Early surgical evacuation of ICH, if performed in a minimally-invasive fashion, may have the potential to minimize secondary injury, reduce edema and mass effect, and improve clinical outcomes”.

These data underline the importance of the detrimental effects of extravascular blood.

The role of red blood cells and their constituents is extremely underrepresented in this review. Basic literature is missing.

  • We thank the reviewer for their feedback, and we have edited the section regarding hemolysate to include the additional references recommended by the reviewer. It is our belief that these edits have improved the quality of this section. Of note, we did not want to greatly expand the size of this section, as the manuscript is already quite large, and we did not want to overwhelm the reader.
  1. For example: the first papers about heme toxicity towards vascular endothelial cells sensitizing them against reactive oxygen species are very important publications opening the field of heme toxicity.
  • We agree that these publications would make a valuable addition. We have added the statement “Secondly, hemin can intercalate into the plasmalemma and sensitizes endothelial cells to ROS mediated damage…” and the associated references from Balla et al from PNAS 1993 and Balla et al from Lab Invest. 1991. This addition is found on line 260.
  1. In this review there is no mention about endogenous, intracellular protective systems against heme stress mediated free radical toxicity, for instance heme oxygenase-ferritin system, where ferritin is the ultimate protectant.
  • We believe that we had previously addressed endogenous degradation systems in the original form of the manuscript. On line 240 we had previously stated “Intracellular hemin is degraded by neuronal and glial heme oxygenase (HO)-1 or -2 in a NADPH-utilizing reaction to biliverdin, carbon monoxide, and free iron [29, 152]. Biliverdin is processed to bilirubin, whereas the free iron is sequestered by ferritin [29, 153], which is finally degraded to inert hemosiderin by lysosomes [29].”
  1. New literatures discuss the toxic heme effect on endoplasmic reticulum and mitochondrial function, they should be included into the paper.
  • We agree that effects on the ER and mitochondria are important to understanding the overall role of heme toxicity. On line 263, we now state: “heme induces endoplasmic reticulum and mitochondrial stress responses” with references from Gall 2018 and Higdon 2012.
  1. The role of HO-1 should be discussed in a more detail form, as it is known as a controversial enzyme in the central nervous system. What is the role of bilirubin, the endproduct of the heme oxygenase-biliverdin reductase enzymes, after brain hemorrhage, is it a protective antioxidant or the agent of bilirubin encephalopathy.
  • This comment is addressed along with the next comment. Please see our response to comment 5.
  1. Carbon monoxide, another endproduct of HO-1 enzyme has multiple effects on cellular metabolism. It would be important to discuss in this article.
  • We appreciate the reviewer’s suggestion, and we agree that a brief discussion of HO and HO end-products would enhance the manuscript. Starting on line 265 we now include a paragraph that addresses points 4 and 5 that begins with “Hemin is degraded by HO into biliverdin, carbon monoxide (CO), and free iron...”
  1. In the field of hemoglobin, it needs to be clarified what forms of hemoglobin the authors mean by oxidized hemoglobin: ferro-hemoglobin with oxygen on it, methemoglobin, ferrylhemoglobin, hemichrome. They have different pathological effects.    
  • On line 234, we now note that there exist multiple different forms of oxidized hemoglobin, and cite the article by Jeney et al., Oxid Med Cell Longev. 2013 for further reading. However, we would prefer to avoid an extended discussion regarding this topic for fear of obscuring the overall message of the manuscript.
  1. Table 1. Heme concentration derives from hemoglobin concentration in the whole blood. Only free heme has clinical relevance. Although the table originates from another paper, it is misleading. Since this not a clear table that is why it could lead to a not correct sentence: „Hemin, the oxidized form of heme, is strongly associated with neuronal death. Blood contains approximately 10 mM of hemin, which is approximately the LD50 for neurons.” This paragraph needs correction.
  • The reviewer raises a reasonable point. We have kept the row containing the concentration of hemoglobin in Table 1, but have removed the rows containing oxyhemoglobin and heme. Additionally, the paragraph in question on line 255 has been altered to reflect this point.
  1. It is not correct, that globin is an innocent player in hemorrhage. For example, peptides derived from oxidized hemoglobin represent proinflammatory effects.
  • The statement “Notably, since globin itself is nontoxic…” has been removed.

Since the review states: “we evaluate the direct role of blood as a neurotoxin“, the above items should implement into this paper. Although the clinical study part of the article is based on uptudate literature, the situation not the same for the hemolysate paragraph.

Reviewer 4 Report

The authors present a review about the role of CNS hemorrhage, comprising intracerebral, intraventricular, and subarachnoid hemorrhage, focusing on the role of extravasated blood.

This contribution represents a relevant, scientific attempt to clarify blood neurotoxicity: as the authors correctly state, in CNS hemorrhage "blood is generally viewed as a basic science curiosity rather than a clinically meaningful factor". 

The authors would be pleased with their work, which goes straight to the point, with an accurate overview of the potential pathogenetic mechanisms

Only one minor issue: in the clinical, daily practice, especially in aSAH, lumbar puncture has a relevant role; there is some study specially focusing on this point?

Author Response

The authors present a review about the role of CNS hemorrhage, comprising intracerebral, intraventricular, and subarachnoid hemorrhage, focusing on the role of extravasated blood.

This contribution represents a relevant, scientific attempt to clarify blood neurotoxicity: as the authors correctly state, in CNS hemorrhage "blood is generally viewed as a basic science curiosity rather than a clinically meaningful factor". 

The authors would be pleased with their work, which goes straight to the point, with an accurate overview of the potential pathogenetic mechanisms

Only one minor issue: in the clinical, daily practice, especially in aSAH, lumbar puncture has a relevant role; there is some study specially focusing on this point?

  • We thank the reviewer for their supportive comments and their interest in our manuscript. With regards to the use of lumbar puncture, it is typically relegated to diagnosis of subarachnoid hemorrhage. While its sensitivity as a diagnostic test has been studied, this role is somewhat outside the scope of our manuscript. In the manuscript, we do address a potential role for lumbar drainage for removing intrathecal blood after aSAH, which may directly address the reviewer’s question. We would direct the reviewer to those references for further reading.

Round 2

Reviewer 3 Report

I appreciate for accepting my suggestions, and since the changes have been made in the manuscript, a think the paper can be published.

Author Response

We appreciate the editor's comment, and agree that this would enhance the message. We have added the above reference, and we now discuss hemoglobin oxidation and modification of the globin moiety starting on line 220. We also now discuss the effects of modified globin molecules on endothelium on line 241.